# Exploring HIV-1 Transmission Dynamics by Combining Phylogenetic Analysis and Infection Timing

**DOI:** 10.3390/v11121096

**Published:** 2019-11-26

**Authors:** Chris Verhofstede, Virginie Mortier, Kenny Dauwe, Steven Callens, Jessika Deblonde, Géraldine Dessilly, Marie-Luce Delforge, Katrien Fransen, André Sasse, Karolien Stoffels, Dominique Van Beckhoven, Fien Vanroye, Dolores Vaira, Ellen Vancutsem, Kristel Van Laethem

**Affiliations:** 1Aids Reference Laboratory, Department of Diagnostic Sciences, Ghent University, 9000 Ghent, Belgium; virginie.mortier@ugent.be (V.M.); Kenny.Dauwe@ugent.be (K.D.); 2Aids Reference Center, Department of Internal Medicine, Ghent University Hospital, 9000 Ghent, Belgium; Steven.Callens@ugent.be; 3Epidemiology of Infectious Diseases Unit, Scientific Institute of Public Health Sciensano, 1050 Brussels, Belgium; Jessika.Deblonde@sciensano.be (J.D.); Andre.Sasse@sciensano.be (A.S.); Dominique.Vanbeckhoven@sciensano.be (D.V.B.); 4Aids Reference Laboratory, Medical Microbiology Unit, Université Catholique de Louvain, 1200 Brussels, Belgium; Geraldine.Dessilly@uclouvain.be; 5Aids Reference Laboratory, Université Libre de Bruxelles, 1050 Brussels, Belgium; Marie-Luce.Delforge@erasme.ulb.ac.be; 6HIV/STD Reference Laboratory, Department of Clinical Sciences, Institute of Tropical Medicine, 2000 Antwerp, Belgium; KFransen@ext.itg.be (K.F.); fvanroye@itg.be (F.V.); 7Aids Reference Laboratory, Centre Hospitalier Universitaire St. Pierre, 1000 Brussels, Belgium; Karolien.Stoffels@lhub-ulb.be; 8Aids Reference Laboratory, Centre Hospitalier Universitaire de Liège, 4000 Liège, Belgium; dvaira@chu.ulg.ac.be; 9Aids Reference Laboratory, Vrije Universiteit Brussel VUB, 1090 Brussels, Belgium; Ellen.Vancutsem@uzbrussel.be; 10Aids Reference Laboratory, University Hospital Leuven, 3000 Leuven, Belgium; Kristel.Vanlaethem@uzleuven.be; 11Rega Institute for Medical Research, Department of Microbiology, Immunology and Transplantation, KU Leuven, 3000 Leuven, Belgium

**Keywords:** human immunodeficiency virus (HIV), transmission dynamics, transmission networks and clusters, molecular epidemiology, phylogenetic analysis, prevention

## Abstract

HIV-1 *pol* sequences obtained through baseline drug resistance testing of patients newly diagnosed between 2013 and 2017 were analyzed for genetic similarity. For 927 patients the information on genetic similarity was combined with demographic data and with information on the recency of infection. Overall, 48.3% of the patients were genetically linked with 11.4% belonging to a pair and 36.9% involved in a cluster of ≥3 members. The percentage of early diagnosed (≤4 months after infection) was 28.6%. Patients of Belgian origin were more frequently involved in transmission clusters (49.7% compared to 15.3%) and diagnosed earlier (37.4% compared to 12.2%) than patients of Sub-Saharan African origin. Of the infections reported to be locally acquired, 69.5% were linked (14.1% paired and 55.4% in a cluster). Equal parts of early and late diagnosed individuals (59.9% and 52.4%, respectively) were involved in clusters. The identification of a genetically linked individual for the majority of locally infected patients suggests a high rate of diagnosis in this population. Diagnosis however is often delayed for >4 months after infection increasing the opportunities for onward transmission. Prevention of local infection should focus on earlier diagnosis and protection of the still uninfected members of sexual networks with human immunodeficiency virus (HIV)-infected members.

## 1. Introduction

The number of new human immunodeficiency virus (HIV) diagnoses in Belgium is falling, with a decrease of 2% in 2017 compared to 2016 and 27.5% compared to 2012. There is wide access to testing and care [1] and since December 2016, the immediate initiation of antiretroviral therapy after diagnosis is recommended for every patient. In 2017, 16,070 HIV-infected persons were in care and 97% of these were on therapy. Of individuals treated for at least six months, 97% attained a viral load of less than 200 copies/mL [2]. Despite these encouraging numbers, 890 new HIV diagnoses were registered in 2017, corresponding to a rate of 7.9 infections per 100,000 individuals, which is well above the average rate in the European Union (6.9 to 6.2 per 100,000) [3]. Nearly half (49%) of all new diagnoses are established in men having sex with men (MSM), mainly of Belgian or European origin, whereas for infections through heterosexual contacts, representing 48% of all new diagnoses, 49% were diagnosed in Sub-Saharan Africans [2].

Laboratory monitoring and drug resistance analysis is centralized in seven Belgian AIDS (acquired immunodeficiency syndrome) Reference Laboratories. Baseline drug resistance testing of newly diagnosed, therapy-naive individuals is standard practice. As evidenced by many studies, the HIV sequence databases that result from baseline resistance testing may provide a wealth of information on the ongoing local spread of HIV [4,5,6,7,8,9,10,11,12]. To what extent the results inferred from these studies reflects the actual transmission dynamics however largely depend on the degree of representativeness of the samples analyzed, the depth of the sampling and the quality of the individual socio-demographic, laboratory and clinical information available, and the relevance of the results is the highest when targeting concentrated and densely sampled epidemics.

Previous phylogenetic analyses of the Belgian HIV epidemic showed that local HIV acquisition is almost exclusively driven by MSM and is concentrated in transmission clusters [13]. However, knowledge on the specific characteristics of those individuals that contribute most to the transmission in these clusters is lacking. There is a general belief that particularly patients with early (acute) infection sustain virus spread because of the high viral loads during this stage of infection [14,15]. Modeling studies [16] and phylodynamic studies [17,18] support this hypothesis.

In order to further characterize the structure of the current Belgian epidemic, extend insight on the drivers of local transmission, and define gaps in prevention, we combined in this study phylogenetic and demographic analyses with information on the stage of infection at diagnosis. The results show that for the majority of individuals infected locally, a potential source partner was found and that a high percentage was involved in a transmission cluster. In these transmission clusters a nearly equal participation of early and late diagnosed individuals was observed. The sexual networks behind the transmission clusters should be the target of prevention focusing on earlier diagnosis and protection of the still uninfected.

## 2. Materials and Methods

### 2.1. Study Population and Infection Timing

HIV-1 *pol* sequences were collected from 1185 treatment-naive patients who received baseline resistance testing after being diagnosed in Belgium in 2014 (*n* = 555) and 2016 (*n* = 630). Demographic data, collected within the framework of mandatory HIV surveillance, were linked to the individual patients using a pseudonymized identifier. For mode of transmission, the categories were MSM, heterosexual, and other (including intravenous drug use, blood transfusion, and perinatal infections). The 26 patients reporting bisexual behavior were all male. They were classified as MSM because male–male contact was presumed to be their most likely source of infection. HIV-1 subtyping was performed using Rega v3 and Comet [19,20]. The subtype was allocated in case of a concordant outcome of both tools and considered as undefined in all other cases. Time since infection was estimated for patients for whom sufficient leftover serum or plasma, collected within one month after diagnosis, was available (*n* = 1033, 87.2%). Infection timing was performed following an algorithm described before, with slight modifications [21]. Patients diagnosed during the pre-seroconversion stage were classified as early diagnosed and patients with a CD4+ T-cell count of 100 or less were classified as late diagnosed. For all other patients, the BED HIV-1 incidence enzyme immunoassay (EIA) and the HIV-1 limiting antigen (LA)g-Avidity EIA (both from Sedia Biosciences Corporation, Portland, OR, USA) were performed. Patients for whom both assays reported ‘recent’, corresponding to a presumed infection ≤4 months before collection of the sample, were considered as ‘early diagnosed’ and patients for whom both assays reported ‘long-term’, corresponding to a presumed infection of more than 4 months before, were referred to as ‘late diagnosed’. Patients with discordant results for both assays were withdrawn from subsequent statistical analyses. Concordant results were obtained for 927 patients (i.e., 89.7% of the patients diagnosed in 2014 or 2016). The characteristics of the 927 study patients were fully comparable with the characteristics of the extended population of patients diagnosed between 2013 and 2017 used for phylogenetic analysis.

### 2.2. Phylogenetic Analysis

HIV-1 *pol* sequences from the study population were supplemented with 1664 HIV-1 *pol* sequences from patients newly diagnosed in 2013 (*n* = 655), 2015 (*n* = 456) or 2017 (*n* = 553). Sequences were generated using Sanger sequencing by either the TruGene HIV-1 genotyping kit (TruGene, Siemens Healthcare Diagnostics, Eschborn, Germany), ViroSeq HIV-1 Genotyping System (Abbott Molecular, Wavre, Belgium) or an in-house protocol. Since the length of the generated sequences slightly differed between protocols, they were trimmed to the 870 nucleotides that were covered by all methods, representing codons 9 to 99 of the protease gene and 41 to 239 of the reverse transcriptase gene. Alignments were composed in BioEdit [22]. Gaps resulting from the insertions of three nucleotides at codon positions 33 or 35 of the protease gene were observed in less than 20 sequences and were removed. Positions associated with drug resistance, detected in 79 patients, were kept after excluding any influence on the phylogenetic tree topology. The phylogenetic tree was constructed using the maximum likelihood (ML) approach implemented in PhyML 3.0 [23] with automatic selection of the best fit evolutionary model of DNA substitution (GTR + G + I) using the Akaike information criterion. Figure 1 shows the phylogenetic tree of the extended population of 2849 patients. Branch support was obtained by the approximate likelihood-ratio test (LRT) [24]. Identification of clustering was performed as described before [13]. Sequences linked with at least one other sequence with an LRT of ≥0.97 and a mean pairwise distance of ≤0.015 were considered as genetically linked and classified as either paired (2 members) or clustered (≥3 members). Sequences located in a cluster but with a branch length of >0.030 were considered as isolated and removed as cluster members to ensure that the cluster members identified primarily resulted from recent transmission events. 

### 2.3. Statistical Analysis 

The relationships between time of infection and patient characteristics, as well as between origin (Belgian versus African) and patient characteristics were modeled using binary logistic regression models. Each potential risk factor was included in the univariate analysis and identified risk factors (*p*-value < 0.05) were selected for multivariate logistic regression analyses using the stepwise method. At first, all risk factors identified in the univariate analysis were included in the multivariate analysis. Identified significant risk factors (*p*-value < 0.05) in this first round were selected for a second round of multivariate analysis and this process was repeated until all remaining risk factors were significant. Only the risk factors that remained as significant after the last analysis were listed in the respective tables. The odds ratio with its 95% confidence interval is provided as a summary statistic. Significance was set at < 0.05. All data were analyzed using SPSS (IBM SPSS Statistics for Windows, Version 23.0. Released 2015; IBM Corp, Armonk, NY, USA). 

### 2.4. Ethics

Ethics approval was obtained for all participating sites with the ethical committee of Ghent University Hospital as the coordinating center (EC ref 2014/0717). All sequence data were compiled and merged with the demographic data and infection time data using pseudonymized identifiers. Only aggregated data were presented.

## 3. Results

### 3.1. Patient Characteristics

The study population consisted of 927 patients, 77.3% were male, 56.4% MSM, 49.9% of Belgian origin, 69.5% reported infection in Belgium, and 49.1% were infected with a subtype B strain. These characteristics fully matched those of the extended series of 2856 patients used to construct the phylogenetic tree and define clustering (77.1% male, 57.4% MSM, 48.2% of Belgian origin, 70.5% infected in Belgium and 46.4% infected with a subtype B strain). A close genetic link in the phylogenetic tree was found for 48.3% of the 927 patients with 11.4% involved in a pair and 36.9% in a transmission cluster. Infection time analysis classified 28.6% of the 927 patients as early diagnosed. The odds of being diagnosed early were significantly higher for clustered and paired individuals, for MSM and for patients reporting infection in Belgium (Table 1).

### 3.2. Origin and Infection Stage at Diagnosis

For the 718 (77.5%) patients for whom the origin was known, 358 (49.9%) were of Belgian and 196 (27.3%) of Sub-Saharan African origin. The odds of being of African origin were significantly higher for females, heterosexuals, younger patients, patients reporting infection in Africa, and patients infected with the non-B subtypes A, C or CRF02_AG (Table 2). Due to the important differences between HIV-infected patients of Belgian and African origin, further analysis of the characteristics associated with early diagnosis was done separately for both groups. Amongst the 358 patients of Belgian origin, 37.4% were diagnosed early. The only characteristic that distinguished early from late diagnosed Belgians was younger age (odds ratio of 0.958 per year of age increase; *p* = 0.001; 95% confidence interval 0.953–0.988). Of the 196 patients of African origin, only 12.2% were diagnosed early. The odds of being diagnosed early were significantly higher for clustered or paired individuals (odds ratio of 9.770; *p <* 0.001; 95% confidence interval 3.489–27.354 for clustered individuals, odds ratio of 4.687; *p* = 0.013; 95% confidence interval 1.383–15.890 for paired individuals).

### 3.3. Position in the Phylogenetic Tree and Time of Diagnosis

Overall, a close genetic counterpart in the phylogenetic tree was found for 448 (48.3%) of the 927 patients. Of the 265 early diagnosed, 67.2% were linked, and of the 662 late diagnosed, 40.8% were linked. When considering only patients of Belgian origin, 71.7% of the early diagnosed and 58.9% of the late diagnosed were linked. Figure 2 shows the distribution of positions in the phylogenetic tree (isolated, paired, clustered) for the different populations. 

A total of 105 clusters with at least one new member belonging to the study population was identified. For the five clusters with the highest increase in new members in 2014 and 2016 (respectively 46, 20, 15, 12, and 10 new members), the ratio of early/late diagnosed individuals (18/28, 10/10, 5/10, 6/6, and 8/2) revealed a nearly equal representation.

### 3.4. Characteristics of the Patients Infected in Belgium

Of the 449 patients for whom information on the most likely location of infection was available, 312 (69.5%) reported being infected in Belgium. Patients infected locally were predominantly male (86.9%), of Belgian origin (75.2%), MSM (71.5%), and infected with HIV-1 subtype B (64.7%). A close link in the phylogenetic tree was found for 69.6% (14.1% paired and 55.4% clustered) and 40.7% were early diagnosed. Older age was found to be significantly associated with being diagnosed late (OR 0.975; *p* = 0.015; 95% confidence interval 0.955–0.995). Figure 2 shows the distribution of positions in the phylogenetic tree for the early and late diagnosed, locally infected, individuals. Only 22.8% of the early diagnosed in Belgium resided on an isolated branch in the tree. These patients were more frequently infected with a subtype A, CRF02_AG, C or D virus (31.0% versus 11.8% in the clustered early diagnosed individuals).

## 4. Discussion

Study of the relationship between HIV sequences from patients of defined geographic regions can provide a unique insight into local epidemics. Combination of this information with demographic data and with information on the time between infection and diagnosis may help to better characterize the patients that were infected recently and thus escaped prevention measures. This information may help to increase the efficacy of prevention measures. Unless a seroconversion is demonstrated however, defining the time between infection and diagnosis is challenging. We used a previously validated algorithm that allows to discriminate between patients diagnosed less than four months after infection (early diagnosed) and patients diagnosed at least four months after infection (late diagnosed) [21]. The algorithm relies on two commercial HIV incidence assays, the Sedia BED HIV-1 EIA and the Sedia HIV-1 LAg-Avidity EIA that respectively assess HIV-1 specific antibody concentration and HIV-1 specific antibody affinity. Both assays have been extensively validated and proven to be reliable although the false prediction of an early infection cannot be fully ruled out [25,26,27]. To minimize the number of false predictions, we therefore included in the study population only patients with a concordant result for both assays. Overall, a high percentage (28.6%) of the study patients were classified as early diagnosed.

As HIV care and laboratory monitoring in Belgium is well centralized and baseline resistance testing is routinely performed for all newly diagnosed and treatment-naive individuals that enter care, we assume that our study population includes the majority of patients first-time diagnosed in our country over the intended period. Analyses for infection time were done for patients diagnosed in 2014 and 2016. The phylogenetic analysis however covered the years 2013 to 2017 to ensure that potential sources diagnosed the same year or the year before or after diagnosis of the study patients could be traced. Based on previous experience, we used stringent criteria to define phylogenetic clustering in order to maximize the chance that linkages identified resulted from recent transmission events [13,28,29]. For nearly half (48.3%) of the study population a close genetic link was found in the phylogenetic tree. This percentage is at the higher end of figures reported for comparable populations [4,30,31,32,33,34,35,36,37]. A one-to-one comparison of these studies is however hampered by differences in time period, depth of sampling, geographical region, degree of heterogeneity of the studied population, and lack of uniformity in methods and definition [38]. In our population for instance, marked differences were observed between the patients of Belgian origin and the patients of African origin. Patients of African origin were more often diagnosed late and more frequently found on isolated branches in the phylogenetic tree. This is in line with the assumption that most Africans diagnosed in Belgium were infected abroad. Information on the time of migration to Belgium is unknown so it cannot be excluded that part of the Africans were infected locally by source partners not diagnosed in the covered time frame. Only a limited number of Africans were diagnosed early. These early diagnosed individuals showed high involvement in pairs and clusters, supporting local transmission and participation in existing sexual networks as reported before [13].

Of the overall 265 early diagnosed patients, 50.9% were members of a cluster, a number that increased to 71.7% in case only individuals of Belgian origin were considered. These percentages are well above the 27% of early diagnosed with a phylogenetic link reported for a comparable study in Paris [33]. Although a close link in the phylogenetic tree is not a direct evidence for one-to-one forward transmission, the findings do indicate that the majority of early diagnosed patients in our study population were infected through contact with sexual networks that include already diagnosed individuals though more thorough studies on cluster dynamics will be needed to support this statement [39]. Only 22.8% (29/265) of the early diagnosed patients reporting infection in Belgium were located on isolated branches in the phylogenetic tree. Interestingly, these individuals showed a high prevalence of African subtypes (A, CRF02_AG, C, and D). This may suggest infection by an African partner not diagnosed in Belgium in the covered time frame. Referral to testing and care is known to be more problematic in migrants [1].

Late diagnosed patients constituted a heterogeneous group, but late diagnosed patients that reported infection in Belgium did not differ in characteristics from the early diagnosed locally infected individuals. Acute infection is generally considered the most critical period for transmission because of the high viral load, lack of awareness, and increased risk behavior around the time of infection. A high representation of recently infected patients in phylogenetic clusters is reported in many studies and considered as an important argument supporting this view [12,34,40,41]. Others however have toned down this opinion or reported on the limitations of phylogenetic analysis to define timing of transmission [15,33,42,43,44]. The observed nearly equal involvement of early and late diagnosed patients in transmission clusters in the study presented (respectively 59.9% and 52.4%) is in-line with the observations of Robineau et al. who also reported a mix of early and late diagnosed individuals in transmission networks [33]. Further investigations are needed to define the effective contribution of each on transmission as well as the most likely time of transmission, as individuals diagnosed late may have transmitted early after infection.

The high number of newly diagnosed individuals that could be linked with a potential source of infection supports the belief that the diagnostic coverage in our country is high, at least for the patients infected locally. It also suggests that an important part of these infections could have been prevented because they occurred through contacts with sexual networks that include HIV diagnosed individuals. The high involvement of patients diagnosed more than four months after infection in these transmission clusters points to a need for more timely diagnosis in order to limit opportunities for transmission. Better understanding of the behavior of individuals in transmission clusters may help to define the reasons for delayed diagnosis. The observation that late diagnosed individuals are of older age is not new but interesting as several research groups have reported age-discrepant transmission, with the selection of older sexual partners being significantly associated with HIV infection in young MSM [32,45,46]. Also, the cross border connections of transmission clusters with other European countries will be worthwhile to examine.

Targeting the sexual networks behind transmission clusters will be key for a more impactful deployment of prevention measures to protect still uninfected network members. In this regard, the Transmission Reduction Intervention Project (TRIP) is a promising initiative [47]. The goal of this initiative is to identify not yet diagnosed infections through partner tracing of the newly diagnosed individuals. The project has been successfully implemented in Ukraine, Greece, and the US [11,48,49,50]. The combination of partner tracing initiatives with real-time phylogenetics and infection time analysis may increase the effectiveness of the approach because it will allow to focus on the most active transmission clusters. Extensive partner tracing and network analysis may also provide opportunities to identify priority candidates for pre-exposure prophylaxis (PrEP) [51]. PrEP was successfully introduced in Belgium mid-2017, after a demonstration project that included 200 participants [52]. The effect of this implementation needs to be followed over the coming years and phylogenetic as well as infection time analysis may be useful tools to do so. 

Important strengths of this study are the nationwide coverage, the dense sampling, the use of very recent data, and the combination of phylogenetic analysis and infection time analysis. A limitation however is that for some variables, such as the presumed location of the infection, the number of missing data is large. Another limitation is that conclusions on transmission were drawn from indirect links, as the phylogenetic methods used do not define direct transmission nor the order of transmission. Moreover, the contribution of undiagnosed individuals, diagnosed individuals without a baseline resistance test, and individuals diagnosed before 2013 or after 2017 could not be assessed. 

Despite these limitations our data revealed some important characteristics of the local HIV epidemic that may offer new opportunities for prevention. Decreasing the time between infection and diagnosis and increasing the awareness of prevention as well as promotion of PrEP and/or condom use in not yet infected partners of sexual networks linked with HIV transmission clusters will be key to further limit local expansion of HIV.

## Figures and Tables

**Figure 1 viruses-11-01096-f001:**
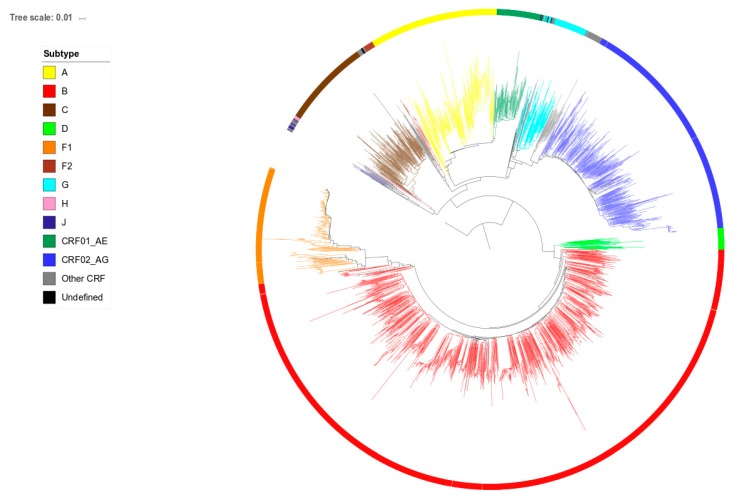
Phylogenetic tree of sequences from human immunodeficiency virus (HIV)-1 infected patients newly diagnosed in Belgium between 2013 and 2017. Tree branches are colored according to subtype. The tree is rooted on a reference subtype J sequence.

**Figure 2 viruses-11-01096-f002:**
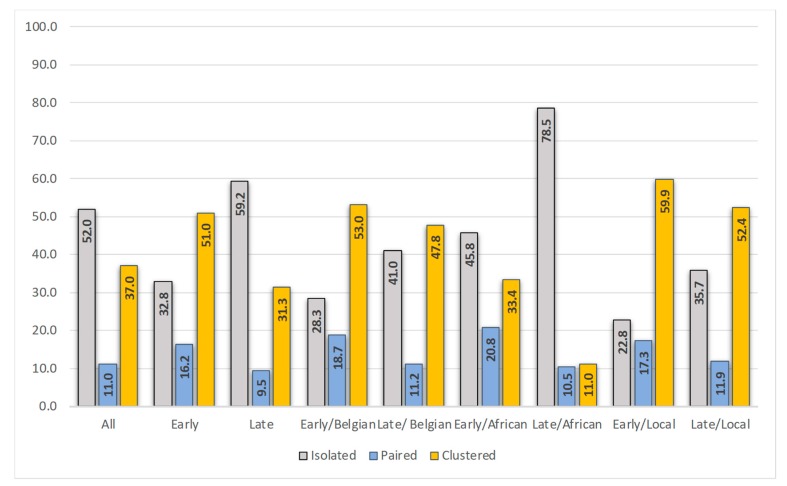
Position in the phylogenetic tree (isolated, paired or clustered, %) for the complete study population (all, *n* = 927), the early diagnosed (*n* = 265), late diagnosed (*n* = 662), early diagnosed of Belgian origin (*n* = 134), late diagnosed of Belgian origin (*n* = 224), early diagnosed of Sub-Saharan African origin (*n* = 24), late diagnosed of Sub-Saharan African origin (*n* = 172), early diagnosed locally infected (*n* = 127), and late diagnosed locally infected (*n* = 185).

**Table 1 viruses-11-01096-t001:** Characteristics associated with being diagnosed early. Results of univariate and multivariate stepwise logistic regression analysis.

		Univariate Analysis			Multivariate Analysis		
	*n*	OR	*p*-Value	95% Confidence Interval	OR	*p*-Value	95% Confidence Interval
Age (per year increase)							
	927	0.991	0.181	(0.979–1.004)			
Year of diagnosis							
2014	437	1.000	-	Ref			
2016	490	1.020	0.893	(0.767–1.357)			
Gender						NS	
Male	717	1.000	-	Ref			
Female	210	0.373	<0.001	(0.248–0.561)			
Position in tree							
Isolated	479	1.000	-	Ref	1.000	-	Ref
Clustered	106	2.939	<0.001	(2.139–4.038)	2.076	<0.001	(1.475–2.921)
Paired	342	3.075	<0.001	(1.957–4.833)	2.65	<0.001	(1.648–4.263)
Transmission risk							
MSM	413	1.000	-	Ref	1.000	-	Ref
Heterosexual	306	0.340	<0.001	(0.240–0.483)	0.486	<0.001	(0.334–0.707)
Other	13	0.127	0.048	(0.016–0.982)	0.224	0.164	(0.027–1.839)
Unknown	195	0.442	<0.001	(0.300–0.653)	0.702	0.116	(0.451–1.091)
Origin						NS	
Belgian	358	1.000	-	Ref			
African	196	0.233	<0.001	(0.145–0.376)			
European	98	0.703	0.153	(0.433–1.140)			
Other	66	0.580	0.071	(0.321–1.048)			
Unknown	209	0.689	0.047	(0.477–0.995)			
Location of infection							
Belgium	312	1.000	-	Ref	1.000	-	Ref
Africa	73	0.062	<0.001	(0.019–0.203)	0.140	0.001	(0.042–0.470)
Europe	32	0.763	0.488	(0.356–1.638)	0.939	0.878	(0.421–2.092)
Other	32	0.662	0.301	(0.303–1.145)	1.241	0.613	(0.538–2.865)
Unknown	478	0.456	<0.001	(0.335–0.621)	0.62	0.008	(0.436–0.881)
Subtype						NS	
B	455	1.000	-	Ref			
02_AG	108	0.732	0.191	(0.458–1.168)			
F	79	1.220	0.430	(0.744–2.001)			
A	78	0.436	0.008	(0.237–0.803)			
C	66	0.315	0.002	(0.152–0.653)			
01_AE	33	0.997	0.993	(0.471–2.109)			
Other/undefined	108	0.453	0.003	(0.269–0.764)			

OR: odds ratio; Ref: used as reference.; NS: non-significant after multivariate analysis; MSM: men having sex with men.

**Table 2 viruses-11-01096-t002:** Characteristics associated with being of African origin (in comparison with being of Belgian origin). Results of univariate and multivariate stepwise logistic regression analysis.

		Univariate Analysis			Multivariate Analysis		
*Risk Factors*	*n*	OR	*p*-Value	95% Confidence Interval	OR	*p*-Value	95% Confidence Interval
Age (per year increase)							
	554	0.958	<0.001	(0.958–0.974)	0.928	<0.001	(0.904–0.953)
Year of diagnosis							
2014	276	1.000	-	Ref			
2016	278	0.899	0.551	(0.635–1.274)			
Infection Stage						NS	
Early	158	1.000	-	Ref			
Late	396	4.287	<0.001	(2.659–6.913)			
Gender							
Male	418	1.000	-	Ref	1.000	-	Ref
Female	136	13.639	<0.001	(8.504–21.876)	2.529	0.013	(1.214–5.266)
Position in tree						NS	
Isolated	273	1.000	-	Ref			
Clustered	73	0.153	<0.001	(0.097–0.241)			
Paired	208	0.418	0.002	(0.242–0.723)			
Transmission risk							
MSM	280	1.000	-	Ref	1.000	-	Ref
Heterosexual	234	11.339	<0.001	(7.221–17.806)	3.414	0.001	(1.710–6.816)
Other	10	18.083	<0.001	(4.452–73.457)	2.885	0.267	(0.445–18.710)
Unknown	30	11.625	<0.001	(5.130–26.341)	6.794	0.001	(2.124–21.728)
Location of infection							
Belgium	245	1.000	-	Ref	1.000	-	Ref
Africa	70	104.963	<0.001	(38.861–283.506)	17.405	<0.001	(5.551–54.574)
Europe	14	0.621	0.652	(0.078–4.936)	0.854	0.894	(0.083–8.755)
Other	14	0.621	0.652	(0.078–4.936)	3.833	0.281	(0.333–44.090)
Not reported	211	7.556	<0.001	(4.664–12.241)	4.799	<0.001	(2.554–9.017)
Subtype							
B	259	1.000	-	Ref	1.000	-	Ref
02_AG	80	21.311	<0.001	(11.303–40.181)	8.875	<0.001	(3.963–19.873)
F	51	1.909	0.144	(0.802–4.546)	1.196	0.729	(0.433–3.301)
A	36	26.678	<0.001	(11.452–62.149)	8.423	<0.001	(2.878–24.648)
C	45	41.043	<0.001	(17.601–95.707)	22.628	<0.001	(7.247–70.653)
01_AE	13	0.855	0.883	(0.106–6.875)	0.564	0.629	(0.055–5.745)
Other/undefined	70	22.387	<0.001	(11.550–43.392)	6.581	<0.001	(2.753–15.728)

OR: odds ratio; Ref: used as reference. NS: non-significant after multivariate analysis.

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
