# Peer review of "Exploring HIV-1 Transmission Dynamics by Combining Phylogenetic Analysis and Infection Timing"

_viruses, 2019, doi:10.3390/v11121096_

Round 1
Reviewer 1 Report
Major comments
It is unclear to me what the main hypothesis, or driver for the analysis, or main finding are. Based on the methods, the recency and clustering analyses are the most novel sources of data, yet their meaning in terms of characterizing the Belgian epidemic seem incomplete. There are vague comments about the epidemic being network-driven but it is not clear what this means. A high proportion of clustered individuals, and a high proportion of recents among diagnosed individuals, mean that Belgium has a low rate of undiagnosed and a rapid diagnosis rate. All epidemics are network-driven, in that tranmissions happen between people linked together ina sexual network. Please elaborate on what you eman by that phrasing in more detal and also be clear form the offset about what you are trying to do in your analysis and what your main findings are. No mention of Belgian 90-90-90? Why have the authors not done multivariable analysis of their data? Many of the predictive variables are confounded.
Minor comments
The paper is well-written and clear. L33 – suggest change of phrasing, because “If only the patients reporting infection in Belgium in Belgium were considered” has a specific meaning (it means they should be considered, which is not what you mean here). Suggest “If we considered only patients reporting infection in Belgium Last line of abstract is a broad claim. Just because lots of patients diagnosed and clustered doesn’t mean that clusters should necessarily be the focus of prevention (see https://www.ncbi.nlm.nih.gov/pmc/articles/PMC5210024/) L64 – I don’t understand “it is concentrated in networks”. Do you mean sexual networks where everyone is connected? 72 what does it mean that transmission is network driven? Do you mean that a small number of individuals are responsible for a large number of onward transmissions? 72 half of individuals in early infection – this is meaningful only as a stat comparison to diagnosis rates as a whole 77 why not 2015? can you explain briefly why for two years (2014 and 2016 you have this enhanced data) and then other years are background? Was this a prospective study? What is the difference between the study population and the rest? 84 not sure this subtype classification make sense. Why bother checking that two programs agree if you then analyze all non-B/ non-Fs together anyway? Are you sure there are no differences between B and F that would make them worth analyzing separately throughout? 117 – I don’t understand how you could end up with a sequence in a cluster with a branch length >0.03 given that the criterion is that pairwise distances should be <0.015. Is it because in a first step you find pairwise distances and then you consider the entire monophyletic clade surrounding those clustered sequences a cluster, then cut out those with long branches? Table 1 – many of these categories will be overlapping/correlated, why have the authors not applied a multivariable model/ (e.g. logistic regression with outcome early/late) Table 2. Same as Table 1. All that you’re saying with these 2 analyses is that you have two groups: early, male, MSM, Belgian, B/F, clustered and late, female, African, heterosexual, African non-B/F, not clustered. I think you need to investigate further as to whether there is anything more interesting going on here. Table 3 goes some way towards this, at least splitting those two groups. Nonetheless, a multivariable approach would be more statistically sound. Table 4, only one significant finding, no need to include table. Just show results for clustering and age and say the rest was not significant. Not really clear what meaning you are attributing to the clustering L225 – and the clustering method L229 – I think this is your most interesting finding. L240 – so is this what you mean when you say people are infected from the network -that most people are infected from transmission networks that include already diagnosed people? L260 – also important to discuss that being diagnosed late doesn’t mean transmissions didn’t happened when individual was in early infection. Differentiation bets stage of infection at diagnosis and transmission? (https://www.ncbi.nlm.nih.gov/pubmed/19133810) 274 border? 287 – highly network driven, I do not understand what you mean
Author Response
Reviewer 1
Major comments
It is unclear to me what the main hypothesis, or driver for the analysis, or main finding are. Based on the methods, the recency and clustering analyses are the most novel sources of data, yet their meaning in terms of characterizing the Belgian epidemic seem incomplete. There are vague comments about the epidemic being network-driven but it is not clear what this means. A high proportion of clustered individuals, and a high proportion of recents among diagnosed individuals, mean that Belgium has a low rate of undiagnosed and a rapid diagnosis rate. All epidemics are network-driven, in that transmissions happen between people linked together in a sexual network. Please elaborate on what you mean by that phrasing in more detail and also be clear form the offset about what you are trying to do in your analysis and what your main findings are.
We understand these concerns and have revised the manuscript thoroughly with special attention for a better description of the study design, the conclusions and suggested actions. We appreciate the suggestions made and have incorporated them in our conclusions. Important modifications were made in the abstract, the last paragraph of the introduction and the discussion section.
No mention of Belgian 90-90-90? Why have the authors not done multivariable analysis of their data? Many of the predictive variables are confounded.
In Belgium the 90-90-90 goal is supposed to be achieved and the target has moved to 95-95-95. But only reliable measurements of the number of diagnosed patients in care and on the number that is treated are available. The number diagnosed is estimated using modelling. In 2017 the number of undiagnosed was estimated to be 10.9% of those infected. We agree with your remark that our results point in the direction of a high diagnostic rate, especially in the native and locally infected population and we have cited that in the discussion.
With regard to the statistical analysis, all comparisons have been redone using multivariate logistic regression analysis. All tables were replaced with new ones. All previous findings were confirmed in the multivariate analysis.
Minor comments
The paper is well-written and clear. L33 – suggest change of phrasing, because “If only the patients reporting infection in Belgium in Belgium were considered” has a specific meaning (it means they should be considered, which is not what you mean here). Suggest “If we considered only patients reporting infection in Belgium
The sentence has been changed.
Last line of abstract is a broad claim.
The abstract has been modified thoroughly including a clearer conclusion.
Just because lots of patients diagnosed and clustered doesn’t mean that clusters should necessarily be the focus of prevention (see https://www.ncbi.nlm.nih.gov/pmc/articles/PMC5210024/).
We are aware of the shortcomings of phylogenetic analysis but attempted to minimize potential errors by using high cut-offs for bootstrap (likelihood ratio test) and pairwise distance. Also, we concentrate on a small geographic region that is extensively covered. We agree that it may be confusing and incorrect to generalize that ‘clusters’ should be the target of prevention and we have removed it from the text. What we really mean is that future prevention initiatives should focus on the sexual networks behind these clusters with as major aim to shorten the time between infection and diagnosis in those infected and protect the still uninfected members. This is made more clear now in the new version of the manuscript.
Al L64 – I don’t understand “it is concentrated in networks”. Do you mean sexual networks where everyone is connected?
We indeed mean sexual networks. This is now clarified throughout the text. See also the remark above.
72 what does it mean that transmission is network driven? Do you mean that a small number of individuals are responsible for a large number of onward transmissions?
We don’t think there is an answer to this question because phylogenetic analysis alone does not allow to discriminate between one by one transmission or presence of one or a few ‘super’-transmitters as most likely cause for the origination and growth of clusters. This information is of high importance and the search for ways on how to obtain it should be the focus of further research.
72 half of individuals in early infection – this is meaningful only as a stat comparison to diagnosis rates as a whole
We do not fully understand this comment and could not trace the sentence where this comment refers to.
77 why not 2015? can you explain briefly why for two years (2014 and 2016 you have this enhanced data) and then other years are background? Was this a prospective study? What is the difference between the study population and the rest?
The study was initially set-up as a retrospective study starting in 2015 with infection time analysis of the patients diagnosed in 2014 that have entered care (baseline sequence available). After analysis of these results and considering the findings as interesting, an extension of the number of patients with data on infection time was decided on and the newly diagnosed patients of 2016 with baseline sequence were additionally analyzed. For practical and financial reasons it was difficult to also include the year 2015 and we don’t think it would have changed the findings.
The patients included in global phylogenetic analysis were patients diagnosed between 2013 and 2017 with available sequence. Demographic data are available for all these patients through our regular surveillance program. We compared the demographic data of the patients diagnosed in 2013, 2015 and 2017 with those diagnosed in 2014 and 2016 and with the overall population diagnosed between 2013 and 2017 and found no differences. We have added the information on this comparison the new version of the manuscript.
84 not sure this subtype classification make sense. Why bother checking that two programs agree if you then analyze all non-B/ non-Fs together anyway? Are you sure there are no differences between B and F that would make them worth analyzing separately throughout?
We agree. In the new statistical analysis the most prevalent subtypes were analyzed separately.
117 – I don’t understand how you could end up with a sequence in a cluster with a branch length >0.03 given that the criterion is that pairwise distances should be <0.015. Is it because in a first step you find pairwise distances and then you consider the entire monophyletic clade surrounding those clustered sequences a cluster, then cut out those with long branches?
We first selected clusters with an LRT of ≥0.97 and then calculate the mean pairwise distance for all members of this clusters to retain only those with a mean pairwise distance of ≤0.015. Because we calculated the ‘overall’ mean pairwise distance of a cluster the presence of single sequences with a branch-length of >0.030 cannot be excluded. We decided to remove these patients as cluster members to increase the probability that the clusters and all cluster members result from relatively recent transmission events. Only three patients were removed because of long branch length.
Table 1 – many of these categories will be overlapping/correlated, why have the authors not applied a multivariable model/ (e.g. logistic regression with outcome early/late) Table 2. Same as Table 1.
Multivariate analysis has been done
All that you’re saying with these 2 analyses is that you have two groups: early, male, MSM, Belgian, B/F, clustered and late, female, African, heterosexual, African non-B/F, not clustered. I think you need to investigate further as to whether there is anything more interesting going on here.
This is what we believe we have done by focusing on the infections that most likely resulted from local transmission.
Table 3 goes some way towards this, at least splitting those two groups. Nonetheless, a multivariable approach would be more statistically sound.
Multivariate analysis has been done.
Table 4, only one significant finding, no need to include table. Just show results for clustering and age and say the rest was not significant.
We agree. The table has been removed. A multivariate analysis has been done for this comparison leading to the same conclusions. Results were included in the text and no longer presented in a table. But the information that was originally provided in table 3 is split over 2 tables in the revised manuscript. That is why there still is a table 4 in the new version.
Not really clear what meaning you are attributing to the clustering
We have tried to specify this more in the revised text.
L225 – and the clustering method
Likewise
L229 – I think this is your most interesting finding.
L240 – so is this what you mean when you say people are infected from the network -that most people are infected from transmission networks that include already diagnosed people?
This is indeed our most important conclusion, that the majority of people infected locally are infected through contact with networks including already diagnosed individuals. These networks are primarily constituted of MSM. This is the known most at risk population in our country and the target population for most of the prevention campaigns that apparently still do not pay off.
These shortcomings of the prevention should be further investigated: why are there so many with delayed diagnosis? What can be done to reach the still uninfected members of sexual networks for protection? We have tried to incorporate these thoughts better in the new version of the manuscript.
L260 – also important to discuss that being diagnosed late doesn’t mean transmissions didn’t happened when individual was in early infection. Differentiation bets stage of infection at diagnosis and transmission? (https://www.ncbi.nlm.nih.gov/pubmed/19133810)
This is definitely true and we have added this consideration to the discussion.
274 border?
This was an awkward spelling mistake. Thanks for noticing.
287 – highly network driven, I do not understand what you mean
Has been removed.
Reviewer 2 Report
The manuscript describes transmissions of HIV within Belgium over a two year period. Overall, the manuscript is well written but some suggestions are listed here that could improve the manuscript.
1) It was not clear to me what the statistical tests were comparing. In the legends to the table it is stated that Bold values were compared, but there are many bold values, and only one p value. Could you please clarify this and present it more clearly.
2) Given that there appear to be linked transmissions and clusters, the data could be presented as such in a figure. [see https://www.sciencedirect.com/science/article/pii/S1755436517301913#fig0010 as an example]
In this way, the reader can visually interpret how transmissions occur, it does not infer a direction, but rather the number of clusters, the number of people involved in each cluster and their connectedness with each other.
3) On line 228 it states "in line with the fact that most of the patients of African origin are most likely infected in Africa resulting in a larger time span between infection and diagnosis". Is there any evidence that the individuals recruited into this study were recent migrants and could have been infected in Africa? Is the alternative hypothesis that they are socially isolated within their racial groups in Belgium and therefore transmissions occur within this group rather than between Anglo Saxon Belgiums and African Belgiums?
Author Response
Reviewer 2
The manuscript describes transmissions of HIV within Belgium over a two year period. Overall, the manuscript is well written but some suggestions are listed here that could improve the manuscript.
1) It was not clear to me what the statistical tests were comparing. In the legends to the table it is stated that Bold values were compared, but there are many bold values, and only one p value. Could you please clarify this and present it more clearly.
For the new version of the manuscript all statistical analyses have been redone using multivariate logistic regression. New tables listing odds ratio’s, p-values and 95% confidence intervals are now provided.
2) Given that there appear to be linked transmissions and clusters, the data could be presented as such in a figure. [see https://www.sciencedirect.com/science/article/pii/S1755436517301913#fig0010 as an example]
In this way, the reader can visually interpret how transmissions occur, it does not infer a direction, but rather the number of clusters, the number of people involved in each cluster and their connectedness with each other.
The suggested analysis and cluster representation is indeed interesting and results in nice visual representations, but we have the feeling that they are not suited for the analyses and the information that we are presenting. We do not focus on the extend or growth of the transmission clusters nor on the inter-cluster relationship and in our opinion it would be incorrect to draw conclusions on this from the data that we have.
3) On line 228 it states "in line with the fact that most of the patients of African origin are most likely infected in Africa resulting in a larger time span between infection and diagnosis". Is there any evidence that the individuals recruited into this study were recent migrants and could have been infected in Africa? Is the alternative hypothesis that they are socially isolated within their racial groups in Belgium and therefore transmissions occur within this group rather than between Anglo Saxon Belgiums and African Belgiums?
Thank you for these thoughts. These are difficult questions to answer as a lot of information on this African population is missing, like time of migration. We have included these shortcomings in the discussion of the new version of the manuscript. We think that most of the infections diagnosed in Africans are imported and resulted from infections in Africa. From previous work we know that there is ample evidence from phylogenetic data for local transmission amongst Africans but indeed, we cannot exclude social isolation, less testing, less referral to care in this population. What we did observe was an intermingling of Belgians and Africans in local MSM clusters.
Reviewer 3 Report
In this manuscript, Verhofstede et al. provide information on the evolution and recent HIV infections in Belgium, by reporting demographic and epidemiological data concerning individuals living in the country of European or African origin. This study is part of the continuous effort of the group to describe the epidemiology of HIV infection in their country. Although some of the reported information is useful for public health policies at a national level, I found the paper to be a mere description of the available information and with little insight and/or novelty considering previous publications of the same group. For example, much of the work reported in the paper is partially overlapping with a previous report of the same group [Verhofstede et al. (2018) Infect Genet Evol, 61, 36].
In addition, the final sentence of the abstract considering “transmission clusters” as primary targets of preventive measures receive little attention in the paper, and cannot be considered a conclusion of this work.
The proportion of missing data is too high in some Tables (e.g. location of infection in Table 1; and Table 2).
Other points:
1) Pol should be italicized throughout the whole manuscript.
2) Many references are incomplete (pages or article numbers missing) (e.g. 12, 20, 21, 26, 28-31, 34, 38, 44, 47
3) Delete “15” after last author name at line 8.
Author Response
Reviewer 3
In this manuscript, Verhofstede et al. provide information on the evolution and recent HIV infections in Belgium, by reporting demographic and epidemiological data concerning individuals living in the country of European or African origin. This study is part of the continuous effort of the group to describe the epidemiology of HIV infection in their country. Although some of the reported information is useful for public health policies at a national level, I found the paper to be a mere description of the available information and with little insight and/or novelty considering previous publications of the same group. For example, much of the work reported in the paper is partially overlapping with a previous report of the same group [Verhofstede et al. (2018) Infect Genet Evol, 61, 36].
In addition, the final sentence of the abstract considering “transmission clusters” as primary targets of preventive measures receive little attention in the paper, and cannot be considered a conclusion of this work.
We understand the comment that the manuscript as presented was mainly descriptive. This was also rightly pointed out by the first reviewer. Therefore we have thoroughly revised the text. Conclusions as well as suggestions for actions and further investigations are now more clearly and more extensively described.
The results of our previous study were indeed the starting point of this study but with this study we tried to focus on the local transmission and on transmissions that escaped the current prevention measures, in particular infections acquired locally and which were diagnosed early.
The proportion of missing data is too high in some Tables (e.g. location of infection in Table 1; and Table 2).
A high number of missing data is inherent to this type of research, using information provided by the patients themselves. Because of the high number of patients included in the study we are convinced however that the number of available data is high enough to support for the conclusions made. The high number of missing data is quoted as a drawback in the discussion section.
Other points:
1) Pol should be italicized throughout the whole manuscript.
This has been corrected.
2) Many references are incomplete (pages or article numbers missing) (e.g. 12, 20, 21, 26, 28-31, 34, 38, 44, 47
This has been corrected. For the online-only journals without page numbering, the article number is provided.
3) Delete “15” after last author name at line 8.
Has been corrected.
Round 2
Reviewer 1 Report
Note – lines refer to those in the version with changes accepted.
L32 – infections reported to be locally acquired?
L100 - when you saw withdrawn from further analysis, you mean you didn’t include those sequences in the phylogeny? This would be a mistake. I think you just mean the stats/demography wasn’t done on those patients so be clear.
L130 & l134 – withheld is the wrong word here, as it means held back/ kept away. I suggest “carried forward” or “selected for”
L137-139. This is exactly the same sequence as l122-125
Table 1 -age and year were not significant in the multivariate? (you have not written NS in that column) Table 1 is hugely improved from previous version of manuscript, results are much more insightful
Table 2 – Results are obvious, maybe include as a supplementary table. I think the results were adequately summarised in the text.
L183 – maybe these results could be better summarised by a histogram or table - % clustering, % pairs, by origin and recency.
Table 3 – Again, I don’t understand why year and gender don’t have results or NS in the multivariate columns. Also, given that nothing is significant in this model, the table should not be in the paper. If you want, it can go in supplementary, but it is enough to summarise in the text that only age was significant.
L240 – and because of differences in methods and definitions https://www.nature.com/articles/srep32251
L248 was should be were
L256 – this isn’t necessarily true. In order to test this, you would have to look for co-clustering of recent infections, see https://journals.plos.org/ploscompbiol/article?id=10.1371/journal.pcbi.1002552
L259 – I am lost with “this small group” which group do you mean?
So, when you look at early patients overall, there is some association between being diagnosed early and being MSM, clustered, from Belgium. When you split into African Belgium, nothing is significant anymore. Does this mean that recency is just a proxy for being diagnosed in Belgium?
Author Response
Dear reviewer,
I would like to express my gratitude for the work you have invested in this review. Your remarks and thoughts have triggered us and brought new insights that have greatly benefited the manuscript and encouraged us.
Thank you on behalf of all authors.
Please find below our reactions to the last remarks
L32 – infections reported to be locally acquired?
Agree. The sentence has been changed as suggested.
L100 - when you saw withdrawn from further analysis, you mean you didn’t include those sequences in the phylogeny? This would be a mistake. I think you just mean the stats/demography wasn’t done on those patients so be clear.
This is correct, they were included in the phylogenetic analysis but not in the statistical analysis. We have changed the sentence to ‘Patients with discordant results for both assays were withdrawn from subsequent statistical analyses’ for better understanding
L130 & l134 – withheld is the wrong word here, as it means held back/ kept away. I suggest “carried forward” or “selected for”
This is indeed a good suggestion. We have changed the wording in the paragraph to:
‘Each potential risk factor was included in the univariate analysis and identified risk factors (p-value <0.05) were selected for multivariate logistic regression analyses using the stepwise method. At first, all risk factors identified in the univariate analysis were included in the multivariate analysis. Identified significant risk factors (p-value <0.05) in this first round were selected for a second round of multivariate analysis and this process was repeated until all remaining risk factors were significant.’
L137-139. This is exactly the same sequence as l122-125
Indeed. This was an error that remained unnoticed. The sentence was removed in L137-139.
Table 1 -age and year were not significant in the multivariate? (you have not written NS in that column) Table 1 is hugely improved from previous version of manuscript, results are much more insightful
We only used NS to identify the results that were not significant after multivariate analysis. Age and year were not included in the multivariate analysis because they turned out as not significant after univariate analysis. For the univariate analyses we preferred to note the exact p-value as it may provide some additional information. For more clarity we have now indicated that NS stands for ‘non-significant after multivariate analysis’.
Table 2 – Results are obvious, maybe include as a supplementary table. I think the results were adequately summarised in the text.
We do feel that this table has some added value. So if the editor agrees we would very much like to keep it as a main table in the paper.
L183 – maybe these results could be better summarised by a histogram or table - % clustering, % pairs, by origin and recency.
This was a good suggestion. We designed and added a figure.
Table 3 – Again, I don’t understand why year and gender don’t have results or NS in the multivariate columns. Also, given that nothing is significant in this model, the table should not be in the paper. If you want, it can go in supplementary, but it is enough to summarise in the text that only age was significant.
Same remark as above for the year and gender. The table was removed. P-value and 95% confidence interval were added in the text.
L240 – and because of differences in methods and definitions https://www.nature.com/articles/srep32251
This is a valuable suggestion. We have added the differences in methods and definitions as an additional hurdle for comparison between studies and cited the Scientific Reports article.
L248 was should be were
Has been changed.
L256 – this isn’t necessarily true. In order to test this, you would have to look for co-clustering of recent infections, see https://journals.plos.org/ploscompbiol/article?id=10.1371/journal.pcbi.1002552
Yes, this is indeed true. We therefore have added a sentence to the discussion to stress that more thorough analysis will be needed to define the position of early infection and the role on transmission. The work of Volz et al. is cited.
L259 – I am lost with “this small group” which group do you mean?
We understand the confusion. With this small group we mean individuals infected locally but found on an isolated branch in the tree.
We have changed the sentence to: ‘Interestingly, these individuals showed a high prevalence of African subtypes (A, CRF02_AG, C and D). This may suggest infection by an African partner not diagnosed in Belgium in the covered time frame’ for better understanding.
So, when you look at early patients overall, there is some association between being diagnosed early and being MSM, clustered, from Belgium. When you split into African Belgium, nothing is significant anymore. Does this mean that recency is just a proxy for being diagnosed in Belgium?
Recency is not a proxy for being diagnosed in Belgium. We have a lot of Africans that are diagnosed for the first time in our country. But recency can be seen as a proxy for being infected in Belgium.
Reviewer 3 Report
Authors have adequately responded to my previous queries and criticisms.
Author Response
Dear reviewer,
Thank you for your time and effort.